# Burden of Respiratory Syncytial Virus Related Acute Lower Respiratory Tract Infection in Hospitalized Thai Children: A 6-Year National Data Analysis

**DOI:** 10.3390/children9121990

**Published:** 2022-12-17

**Authors:** Phanthila Sitthikarnkha, Rattapon Uppala, Sirapoom Niamsanit, Sumitr Sutra, Kaewjai Thepsuthammarat, Leelawadee Techasatian, Watit Niyomkarn, Jamaree Teeratakulpisarn

**Affiliations:** 1Department of Pediatrics, Faculty of Medicine, Khon Kaen University, 123 Mittraphap Road, Muang, Khon Kaen 40002, Thailand; 2Clinical Epidemiology Unit, Faculty of Medicine, Khon Kaen University, Khon Kaen 40002, Thailand; 3Department of Pediatrics, Faculty of Medicine, Chulalongkorn University, 1873 Rama IV Road, Bangkok 10330, Thailand

**Keywords:** respiratory syncytial virus, respiratory tract infection, children, mortality

## Abstract

Objectives: This study sought to determine the epidemiology, seasonal variations, morbidity, and mortality of respiratory syncytial virus (RSV) infection among hospitalized children with lower respiratory tract infection in Thailand. In addition, we assessed the risk factors associated with severe RSV lower respiratory tract infection (LRTI)-related morbidity and mortality. Methods: The data were reviewed retrospectively from the National Health Security Office for hospitalized children younger than 18 years old diagnosed with RSV-related LRTI in Thailand, between the fiscal years of 2015 to 2020. The RSV-related LRTIs were identified using the International Statistical Classification of Diseases and Related Health Problems, 10th Revision, Thai Modification. ICD-10-TM codes J12.1, J20.5, and J21.0, which represent respiratory syncytial virus pneumonia, acute bronchitis due to respiratory syncytial virus, and acute bronchiolitis due to respiratory syncytial virus, respectively, were studied. Results: During the study period, RSV-related LRTI accounted for 19,340 of the 1,610,160 hospital admissions due to LRTI. RSV pneumonia was the leading cause of hospitalization (13,684/19,340; 70.76%), followed by bronchiolitis (2849/19,340; 14.73%) and bronchitis (2807/19,340; 14.51%), respectively. The highest peak incidence of 73.55 percent occurred during Thailand’s rainy season, from August to October. The mortality rate of RSV-related LRTI in infants younger than 1 year of age was 1.75 per 100,000 person years, which was significantly higher than that of children 1 to younger than 5 years old and children 5 to younger than 18 years old (0.21 per 100,000 person years and 0.01 per 100,000 person years, respectively, *p*-value < 0.001). Factors associated with mortality were congenital heart disease, hematologic malignancy, malnutrition, and neurological disease. Conclusions: In children with RSV LRTI, pneumonia was the leading cause of hospitalization. The admission rate was highest during the rainy season. Mortality from RSV-related LRTI was higher in children under 1 year old and in children with underlying illnesses; future preventive interventions should target these groups of patients.

## 1. Introduction

Respiratory syncytial virus (RSV) is a species of genus Pneumovirus, subfamily Pneumovirinae, family Paramyxoviridae, and order Mononegavirales. Human RSV exists as two antigenic subgroups, A and B [1]. It is the most common cause of bronchiolitis in infants and young children [2]. Hospitalizations of infants due to lower respiratory tract infection (LRTI) are primarily due to RSV [3]. RSV-associated LRTI hospitalizations occur worldwide, with children under the age of 6 months accounting for 45% of hospital admissions and mortality [4]. The illness burden associated with RSV in children is estimated to be an average of 20.8 million cases, 1.8 million hospitalizations, and 40,000 fatalities across 72 countries [5]. RSV-positive children are more likely to have a severe condition, as evidenced by their increased likelihood of hospitalization, intensive care unit (ICU) admission, and oxygen requirements, compared to children infected with other viruses [6]. 

Between 2003 and 2014, the Thailand Ministry of Public Health (MoPH) and the Centers for Disease Control and Prevention (CDC) of Thailand conducted surveillance for LRTI among hospitalized cases in two Thai provinces. As a result of this surveillance, pathogens such as influenza virus (8.2%) and RSV (19.5%) were shown to be predominant among cases of LRTI in children [7]. A study conducted in Thailand during the prior decade found that 25% of hospitalized children under the age of five were admitted with LRTI, although no organism-specific illness was reported [8]. Due to a failure to identify the organisms causing LRTI among Thai children, the burden of RSV diseases among children in Thailand has not yet been documented on a national scale, and so the incidence of this infection was underreported in a previous study [8]. In the majority of Thai hospitals, RSV infection detection has been regularly tested over the last decade. This study was conducted to investigate the frequency, seasonal variation, morbidity, and mortality of RSV-related LRTI in hospitalized children from 2015 to 2020. In addition, we assessed the risk factors associated with severe RSV LRTI-related morbidity and mortality.

## 2. Materials and Methods

### 2.1. Study Design and Participants

We reviewed the data of hospitalized patients under 18 years of age based on the Universal Coverage (UC) scheme in Thailand. The UC scheme is the main social health insurance program in Thailand, covering approximately 72% of the Thai population [9]. All public hospitals and some private hospitals are covered by the UC scheme. The public hospitals are responsible for providing care through three levels, primary, secondary, and tertiary hospitals. We extracted hospital discharge data from the National Health Security Office using the International Statistical Classification of Diseases and Related Health Problems, 10th Revision, Thai Modification: ICD-10-TM of J12.1, J20.5, and J21.0, which account for respiratory syncytial virus pneumonia, acute bronchitis due to respiratory syncytial virus and acute bronchiolitis due to respiratory syncytial virus, respectively. RSV was diagnosed as a cause of LRTI when respiratory tract specimens were detected from molecular diagnosis. The data were retrospectively reviewed from the fiscal years 2015 to 2020. The fiscal year of Thailand ranges from 1 October to 30 September of the following year. (Fiscal year 2015 indicates 1 October 2014 to 30 September 2015).

### 2.2. Data Collections

Demographic data included the patient’s age, gender, month and year of admission, and level of hospitalization. The outcome of RSV-related LRTI was evaluated based on the number of respiratory failures that required endotracheal intubation, the hospital length of stay (LOS), and the mortality rate. The rate of respiratory failure that required endotracheal intubation was identified using The International Classification of Diseases, Ninth Revision, Clinical Modification (ICD-9-CM) of 96.04, 96.70, 96.71, and 96.72, which refer to the insertion of endotracheal tube, continuous invasive mechanical ventilation of unspecified duration, continuous invasive mechanical ventilation for less than 96 consecutive hours, and continuous invasive mechanical ventilation for 96 consecutive hours or more, respectively. We also collected comorbidities of all participants from ICD-10-TM and included this as a binary variable. Comorbidities in this study included conditions such as congenital heart disease, chronic respiratory disease, neuromuscular disease, hematologic malignancy, and malnutrition. The institutional review board of Khon Kaen University approved this study on 27 June 2022 (#HE 641388).

### 2.3. Statistical Analyses

All statistical analyses were performed using the Stata software version 10 (StataCorp. 2007. Stata Statistical Software: Release 10. College Station, TX, USA: StataCorp LP.). We presented number of admissions as episodes and percentage. The admission rate was expressed per 1000 members of the population by year and age. The population estimate data were extracted from official statistic registration systems, including The Bureau of Registration Administration, Department of Provincial Administration, Ministry of Interior, Thailand. The demographic characteristics of patients were described depending on the type of data. The mortality rate was calculated and presented as per 100,000 population of the same age groups. We divided patients into three age groups (under one, one to under five, and five to under eighteen years of age) to evaluate the differences in diagnosis, admission rate, and mortality rate. The univariate and backward stepwise multivariable logistic regression analysis was applied to determine the factors associated with respiratory failure that required endotracheal intubation and mortality. The risk factors for adjusting the model were chosen based on previous literature and clinical experiences, including age, chronic respiratory disease, congenital heart diseases, hematologic malignancy, malnutrition, and neurologic disease. The 95% confidence interval (CI) of the rate was computed based on the normal approximation to the binomial distribution. *p*-values < 0.05 were considered to indicate statistical significance.

## 3. Results

### 3.1. Demographics Data

From 1,610,160 hospital admissions due to LRTI during the fiscal years 2015 to 2020, RSV-related LRTI accounted for 19,340 admissions (1.2%). Among the patients hospitalized due to RSV-related LRTI, more than half were male (11,142/19,340; 57.61%). The most prevalent diagnosis was RSV pneumonia (13,684/19,340; 70.76%), followed by RSV bronchiolitis (2849/19,340; 14.73%) and RSV bronchitis (2807/19,340; 14.51%), respectively. RSV pneumonia had the highest admission rate (0.17 per 1000 person-year). Tertiary care hospitals had the highest rate of RSV-related LRTI hospitalization (10,620/19,340; 54.91%), followed by secondary (27.26%), private (11.66%), and primary care hospitals (6.17%). The proportion of RSV LRTI in hospitalized children divided by principal diagnosis is presented in Table 1. 

Among the children hospitalized with RSV LRTI, 49.54% (9581/19,340) were aged one to under five years, 48.27% (9335/19,340) were under one year of age, and 2.19% were aged five to under eighteen years (424/19,340). The RSV-related LRTI incidence in infants under one year of age, one to under five, and five to under eighteen years accounted for 2.60, 0.58, and 0.01 per 1000 person years, respectively (Table 2). 

### 3.2. Annual Trend of Admission

The annual admission number of children due to RSV-related LRTI from fiscal years 2015 to 2020 is shown in Table 3. 

The incidence of RSV-related LRTI increased from the fiscal years of 2015 to 2018, when the number of RSV-related LRTI admissions peaked. However, it gradually decreased from the fiscal years of 2019 to 2020. RSV-related LRTI hospitalization occurred mainly during August to October, accounting for 73.55% of hospitalizations (14,224/19,340). The lowest rate of hospitalization was seen in the summer (March to May). This monthly trend of admission from RSV-related LRTI had a similar pattern every year during the fiscal years of 2015–2020 (Figure 1).

In comparing the admission rate during the fiscal years of 2015 to 2019 to that during the fiscal year 2020, the admission rates in October 2019 to March 2020 were higher than those in the fiscal years of 2015 to 2019. Nevertheless, admission rates from April to September 2020 were lower compared with the same period in the fiscal years of 2015 to 2019 (Figure 2). 

### 3.3. Outcome of Treatment

Each diagnosis of RSV-related LRTI in hospitalized children led to a different length of stay. RSV pneumonia accounted for the longest length of stay (7.9 ± 18.5 days), followed by RSV bronchitis (4.4 ± 5.8 days) and RSV bronchiolitis (4.1 ± 4.3 days), consecutively.

### 3.4. Respiratory Failure Required Intubation 

There were 1331 cases of acute respiratory failure necessitating intubation out of a total of 19,340 RSV LRTIs (6.88%), with the highest prevalence occurring in children with RSV pneumonia, which accounted for 1257 cases (94.44%). Children under one year old accounted for 66.94% (891/1331) of RSV-LRTI-related respiratory failure cases requiring endotracheal intubation. The fraction of endotracheal intubation was higher in children aged one to five years (398/1331; 29.9%) than in children aged five to ten years (42/1331; 3.16%). Chronic respiratory disease (adjusted odds ratio (aOR) 1.67; 95% CI 1.20–2.32, *p* = 0.002), congenital heart disease (aOR 5.19; 95% CI 4.22–6.40, *p* < 0.001), malnutrition (aOR 3.02; 95% CI 1.99–4.59, *p* < 0.001), and neurological disease (aOR 2.57; 95% CI 1.55–4.29, *p* < 0.001) were the most prevalent comorbidities of respiratory failure requiring endotracheal intubation among children with RSV-related LRTI (Table 4). 

### 3.5. RSV-Related LRTI Mortality 

During the study period, there were 106 cases of hospital mortality related to RSV-associated LRTI, accounting for 0.55% of all hospitalized children with RSV LRTI. Mortality was more prevalent in children under the age of one year, being detected in 65 infants, than in children aged one to five years (0.36%) (Table 2). Factors associated with death were related to underlying medical conditions, including congenital heart disease (aOR 10.09; 95% CI 6.21–16.37, *p* < 0.001), hematologic malignancy (aOR 10.34; 95% CI 4.17–25.63, *p* < 0.001), and neurological disease (aOR 3.50; 95% CI 1.13–10.89, *p* = 0.030) (Table 5).

## 4. Discussion

This nationwide, data-based study offers information on the primary social health insurance system in Thailand (UC scheme), which covers roughly 72 percent of the population [9]. Thus, our data reflect the vast majority of Thai children. RSV-related LRTI poses a huge global public health burden. However, the prevalence of RSV-related lower respiratory tract infections (LRTIs) in children varies greatly among countries based on geography and the diagnostic approaches employed to identify RSV. This study provided additional information regarding RSV-related LRTI, supplementing the prior national study. During the fiscal years of 2015–2020, 1.2% of hospitalized children in Thailand had RSV LRTI. In Thailand, the incidence of RSV-related LRTI grew gradually from 2015 to 2018, possibly due to the advancements in diagnostic techniques during the past decade. The year 2018 saw the highest RSV-related LRTI admission rate ever recorded (0.43 per 1000 population). Nonetheless, after the beginning of the coronavirus disease 2019 (COVID-19) pandemic in March 2020, the rate of RSV-related LRTI hospitalization fell significantly. This decline in the admission rate of infants in Thailand with RSV-related LRTI in 2020 paralleled the trend seen globally [10]. This may have been the result of the wide adoption of public health policies following the start of the pandemic, including the use of non-pharmaceutical treatments (NPIs), such as the use of face masks, hand washing, social distancing, and school closures [11].

Although RSV can infect individuals of all ages, it is the major cause of LRTI hospitalizations in children younger than five years old [12]. This national survey also discovered a high burden of RSV-related LRTI hospitalization among children younger than five years old. Among infants younger than one year of age, the admission rate was 2.60 per 1000 person years, compared to 0.58 per 1000 person years among those older than one to five years. During 2008–2011, the rate of RSV-related LRTI hospitalization among children younger than five years old was 15.43 per 1000 person years, according to population-based monitoring in two rural districts of Thailand [13]. This prior study was an RSV surveillance study that covered all hospitalized children with LRTI; therefore, we cannot directly compare the findings. Thailand is classified as an upper-middle-income country by the World Bank [14]. Compared to countries in the same World Bank income classification, the admission rate of RSV-related LRTI in Thailand was lower than the estimated incidence of countries in the same World Bank income classification, which ranges from 10.2 to 34.5 per 1000 person-year in infants less than 1 year [15].

RSV seasons vary around the world and depend on altitude, climate, and geographic location [16]. This study determined that the peak admission rate for RSV-related LRTI cases in Thailand occurred during the late rainy months (August to October). This seasonal pattern is similar to those discovered by an earlier study in Thailand [17]. Several studies in tropical countries, especially in South East Asia, such as Vietnam [18], the Philippines [19], and Malaysia [20], also reported positive associations between rainfall and RSV-related LRTI hospitalization. This could be explained by a change in human behavior during the rainy season. Children spend more time indoors in both their homes and at school, leading to crowded settings. This environment could influence RSV transmission, as RSV is transmitted through direct and indirect contact [21]. The wet conditions during rainy seasons also increase the amount of virus that is deposited on surfaces, and virus survival in droplets on surfaces encourages transmission of RSV via contact [22]. This annual peak and association with rainfall in Thailand can guide future measures to prevent the spread of RSV infection.

RSV infection can result in a range of clinical manifestations. The symptoms of RSV range from moderate, such as upper respiratory tract infection, to severe, causing respiratory failure, such as bronchiolitis and pneumonia. RSV pneumonia can result in respiratory distress and respiratory failure, necessitating endotracheal intubation [12]. This study discovered that 6.88% of children with RSV-related LRTI required intubation, with the vast majority being diagnosed with pneumonia. Previous research from a tertiary care institution in Thailand indicated that 10% of hospitalized children with community-acquired RSV-related LRTI required intubation and mechanical ventilation [17]. The increased intubation rate in the previous study may have resulted from the underlying condition of patients in tertiary care centers, as the majority of severe cases in Thailand are admitted to tertiary care centers. In addition, the prevalence of chronic lung disease, congenital heart disease, malnutrition, and neurological impairment was found to be related with severe RSV-related LRTI, necessitating mechanical ventilation in our study. In a previous study, neuromuscular disease was independently linked with mechanical ventilation in infants with RSV infection (OR, 3.85; 95% CI, 1.28–10.01; *p*-value = 0.017) [23].

The overall mortality rate in this study was 0.55%, which is comparable to the rate found in an earlier study conducted in Thailand from 2008 to 2011 (0.7%) [13]. Our study found that the majority of deaths occurred in hospitalized children with RSV pneumonia, a finding which is in agreement with a study conducted in Spain [24]. Infants in their first year of life had the highest fatality rate from RSV-related LRTI in this study, accounting for 0.69% of cases. According to a recent comprehensive analysis of the worldwide burden among children of the same age group, this value is similar to the mortality rate in other upper middle-income nations (0.8%, range 0.3–1.9%) [15]. Nonetheless, the mortality rate from RSV-related LRTI among children was higher in low- and middle-income countries than in high-income countries. This could be due to a lack of a humanized monoclonal antibody that has been found to lessen the severity of RSV infection in infants. However, these antibodies are expensive and not yet accessible in low- and middle-income nations such as Thailand.

This study had several limitations. First, it was a retrospective review of the records derived from the ICD-10-TM diagnosis classification, which may have caused an underestimation of RSV-related LRTI, since there could have been some coder diagnosis misclassifications. Second, identification of the organisms causing ALRI among Thai children is underreported because laboratory diagnosis for RSV still limited in several hospitals, especially in primary care hospitals. Third, we only extracted hospital discharge data; hence, clinical manifestations, chest radiograph pattern, and laboratory investigations other than causative organisms were not included. Fourth, we only evaluated the risk factor for intubation and mortality in RSV-related LRTI children. Possible risk factors for the development of RSV LRTI, such preterm birth, or low birth weight, were not included in our study. Fifth, the data for the analysis were derived from the public health ministry’s universal coverage program, which covers 47.8 million people, or 72%, of the population. However, we do not have statistics on the remaining 28%, which is fully reimbursable through the government system (finance ministry and labor ministry) [9].

The findings of this study will assist clinicians in identifying children with risk characteristics who are infected with RSV in the lower respiratory tract. A vaccine against RSV was not available until recently. Consequently, the American Academy of Pediatrics advises high-risk newborns and children receive immunological prophylaxis during RSV season [25]. This palivizumab immune prophylaxis was successful in lowering the incidence of severe disease and hospitalization due to RSV-related LRTI in high-risk babies and children [26]. Due to its high price, however, it has been unavailable in Thailand. Future prospects call for the implementation of national strategies for developing prevention and promotion policies, particularly in high-risk populations for severe RSV disease, such as infants younger than one year of age and children with chronic respiratory disease, congenital heart disease, and neurologic disease. Corresponding to the seasonal variation of RSV, Thailand’s preventive measures must be implemented prior to the August outbreak. Future research should concentrate on developing a vaccination for RSV to prevent infection and antiviral drugs to treat RSV-associated severe LRTI.

## 5. Conclusions

The burden of RSV pneumonia has resulted in the highest admission rate seen to date, particularly among infants less than one year of age. RSV continues to be a substantial cause of morbidity and mortality in children, particularly among infants that are under one year of age and those with comorbidities as well as potential factors that may aggravate clinical outcomes in children with RSV infections. Such factors include immature immune systems, hereditary disorders (cystic fibrosis), and upper respiratory tract deformities (choanal atresia, pyriform aperture stenosis, etc.). Effective preventive measures should primarily target children with risk factors; it is a matter of public health priorities.

## Figures and Tables

**Figure 1 children-09-01990-f001:**
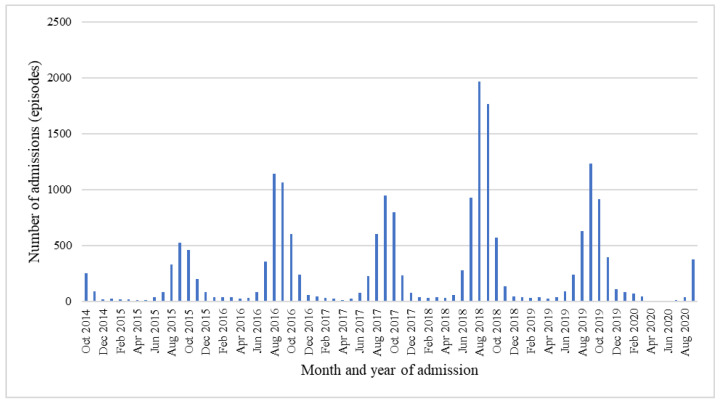
Monthly climate trends of RSV-related LRTI admissions in children from fiscal years 2015 to 2020.

**Figure 2 children-09-01990-f002:**
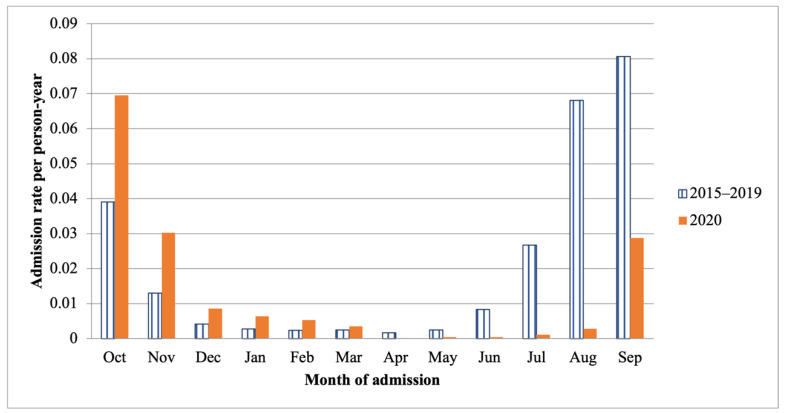
Comparing the admission rate per person year for children with RSV-related LRTI between the fiscal years 2015 to 2019 and the fiscal year 2020.

**Table 1 children-09-01990-t001:** The proportion of RSV LRTI in hospitalized children during the fiscal years 2015–2020.

ICD-10 TM	J12.1	J21.0	J20.5	Total
Principal Diagnosis	RSV Pneumonia (*N* = 13,684)	Acute Bronchiolitis Due to RSV (*N* = 2849)	Acute Bronchitis Due to RSV (*N* = 2807)	(*N* = 19,340)
Fiscal year, No. (%)				
2015	969 (7.08)	187 (6.66)	283 (9.93)	1439 (7.44)
2016	2499 (18.26)	514 (18.31)	558 (19.59)	3571 (18.46)
2017	2062 (15.07)	385 (13.72)	454 (15.94)	2901 (15.00)
2018	4460 (32.59)	964 (34.34)	822 (28.85)	6246 (32.30)
2019	2196 (16.05)	474 (16.89)	452 (15.87)	3122 (16.14)
2020	1498 (10.95)	283 (10.08)	280 (9.83)	2061 (10.66)
Gender, No. (%)				
Male	7854 (57.4)	1573 (56.04)	1715 (61.1)	11,142 (57.61)
Female	5830 (42.6)	1276 (43.96)	1092 (38.9)	8198 (42.39)
Age group, No. (%)				
<1 year	6798 (49.68)	1056 (37.62)	1481 (51.98)	9335 (48.27)
1-<5 years	6641 (48.53)	1622 (57.78)	1318 (46.26)	9581 (49.54)
5-<18 years	245 (1.79)	129 (4.6)	50 (1.76)	424 (2.19)
Level of hospital, No. (%)				
Primary	889 (6.5)	187 (6.66)	117 (4.11)	1193 (6.17)
Secondary	2944 (21.51)	865 (30.82)	1462 (51.32)	5271 (27.26)
Tertiary	8368 (61.15)	1134 (40.4)	1118 (39.24)	10,620 (54.91)
Private	1483 (10.84)	621 (22.12)	152 (5.34)	2256 (11.66)

**Table 2 children-09-01990-t002:** Admission and mortality rates of RSV LRTI by age group in fiscal years 2015–2020.

Age Group (Years)	Admission		Mortality	
	Number	Admission Rate (/1000 Person Time)	Number	Mortality Rate (/100,000 Person Time)
<1	9335	2.60	65	1.75
1–<5	9581	0.58	35	0.21
5–<18	424	0.01	6	0.01

**Table 3 children-09-01990-t003:** Admissions of Thai children due to RSV-related LRTI in fiscal years 2015 to 2020.

Fiscal Year	Number of Admissions	Admission Rate /1000 People
2015	1439	0.10
2016	3571	0.26
2017	2901	0.21
2018	6246	0.46
2019	3122	0.23
2020	2061	0.16

**Table 4 children-09-01990-t004:** Factors associated with respiratory failure requiring endotracheal intubation among hospitalized children with RSV-related LRTI.

	Number of Intubations N, (%)	Univariate			Multivariable		
		Crude OR	95%CI	*p*-Value	Adjusted OR	95%CI	*p*-Value
Chronic respiratory disease							
No	1283 (96.39)	1	-		1	-	
Yes	48 (3.61)	1.56	1.15–2.12	0.004	1.67	1.20–2.32	0.002
Congenital heart disease							
No	1130 (84.90)	1	-		1	-	
Yes	201 (15.10)	8.44	7.01–10.17	<0.001	5.19	4.22–6.40	<0.001
Hematologic malignancy							
No	1318 (99.02)	1	-		1	-	
Yes	13 (0.08)	1.82	1.02–3.27	0.044	1.31	0.70–2.46	0.397
Malnutrition							
No	1279 (96.09)	1	-		1	-	
Yes	52 (3.91)	7.01	4.99–9.88	<0.001	3.02	1.99–4.59	<0.001
Neurological disease							
No	1302 (97.82)	1	-		1	-	
Yes	29 (2.18)	4.16	2.73–6.34	<0.001	2.57	1.55–4.29	<0.001

Abbreviations: 95% CI, 95% confidence interval; OR, odds ratio.

**Table 5 children-09-01990-t005:** Factors associated with mortality among hospitalized children with RSV-related LRTI.

	Number of Deaths N, (%)	Univariate			Multivariable		
		Crude OR	95% CI	*p*-Value	Adjusted OR	95% CI	*p*-Value
Congenital heart disease							
No	70 (66.04)	1	-		1	-	
Yes	36 (33.96)	17.89	11.86–27.00	<0.001	10.09	6.21–16.37	<0.001
Hematologic malignancy							
No	98 (92.45)	1	-		1	-	
Yes	8 (7.55)	15.30	7.24–32.33	<0.001	10.34	4.17–25.63	<0.001
Malnutrition							
No	100 (94.34)	1	-		1	-	
Yes	6 (5.66)	7.63	3.29–17.67	<0.001	1.52	0.54–4.32	0.428
Neurological disease							
No	102 (96.23)	1	-		1	-	
Yes	4 (3.77)	6.19	2.24–17.10	<0.001	3.50	1.13–10.89	0.030

Abbreviations: 95% CI, 95% confidence interval; OR, odds ratio.

## Data Availability

The datasets generated and/or analyzed during the current study are not publicly available but are available from the corresponding authors (RU) upon request.

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
