# Peer review of "Burden of Respiratory Syncytial Virus Related Acute Lower Respiratory Tract Infection in Hospitalized Thai Children: A 6-Year National Data Analysis"

_children, 2022, doi:10.3390/children9121990_

Round 1

Reviewer 1 Report

In the manuscript, the authors have examined the burden of RSV infections nationwide after the RSV infection detection test kit was largely available in the hospitals in Thailand. As described in the manuscript, RSV infections are one of the leading cause of mortality especially among children under 5, and therefore, the descriptive data analysis at the nation level is quite meaningful. As far as I know and the authors stated, the manuscript is the first study to look at the 2015-2020 RSV infection data using the national-level hospitalization data in Thailand. In sum, this paper (and its motivation) is very significant.

Methods and results: The study design and data collections look fine. Although the statistical analyses look fine, the stata version that they used (ver.10) is outdated (released back in 2007). Therefore, the authors may need to use an updated version of Stata (e.g. 16 or 17) and to confirm the results.

Discussion: Although the abstract mentions the role of rainy season (Aug to Oct) in RSV infections, why the rainy season matter has not yet fully explained. One sentence is provided “It can explain from temperature and humidity are influence the epidemic pattern of RSV [19].” but the readers would expect a full one paragraph to discuss the role of temperature and humidity of the Thai rainy season and how to control rainy-season-related RSV case hikes.

Overall, the manuscript is well written but some sentences are not grammatically correct, so, at least the authors or a third party improve English before or after acceptance.

Author Response

Dear Editors and Reviewer

Thank you for taking your time reviewing our manuscript. Your comments are very valuable for improving our writing. We are glad to receive all your valuable comments. Therefore, the authors have discussed, looked back, and edited the manuscript according to the constructive feedback of the manuscript. We really hope that our revision will match the criteria for publication in Children. Our responses to editors are described as follow.

Reviewer # 1

In the manuscript, the authors have examined the burden of RSV infections nationwide after the RSV infection detection test kit was largely available in the hospitals in Thailand. As described in the manuscript, RSV infections are one of the leading cause of mortality especially among children under 5, and therefore, the descriptive data analysis at the nation level is quite meaningful. As far as I know and the authors stated, the manuscript is the first study to look at the 2015-2020 RSV infection data using the national-level hospitalization data in Thailand. In sum, this paper (and its motivation) is very significant.

Methods and results: The study design and data collections look fine. Although the statistical analyses look fine, the stata version that they used (ver.10) is outdated (released back in 2007). Therefore, the authors may need to use an updated version of Stata (e.g. 16 or 17) and to confirm the results.

Response: Your comment is much appreciated. We used Stata version 16 and confirmed that the result was unaltered. Unfortunately, our university's official license is Stata version 10. As a result, Stata version 16 cannot be mentioned in the manuscript.

Discussion: Although the abstract mentions the role of rainy season (Aug to Oct) in RSV infections, why the rainy season matter has not yet fully explained. One sentence is provided “It can explain from temperature and humidity are influence the epidemic pattern of RSV [19].” but the readers would expect a full one paragraph to discuss the role of temperature and humidity of the Thai rainy season and how to control rainy-season-related RSV case hikes.

Response: Thank you very much for your feedback. We rewrite the sentence to “This could be explained by a change in human behavior during the rainy season. Children spend more time indoors in both their homes and at school, leading to crowded settings. This environment could influence RSV transmission, as RSV is transmitted through direct and indirect contact [21]. The wet conditions during rainy seasons also increase the amount of virus that is deposited on surfaces, and virus survival in droplets on surfaces encourages transmission of RSV via contact [22]. This annual peak and association with rainfall in Thailand can guide future measures to prevent the spread of RSV infection.” as shown in line 236-243.

Overall, the manuscript is well written but some sentences are not grammatically correct, so, at least the authors or a third party improve English before or after acceptance.

Response: Thank you very much for your valuable suggestion. This revision we put out for expert English editing service of MDPI as the certification stated below.

Reviewer 2 Report

Globally, in South Global, there have been major reductions in bacterial respiratory infection in under-5, so it became clear how urgent the investments in the prevention, diagnosis and management of children's respiratory infections are. So surveillance of these viral diseases is needed, and regular and timely analysis should be conducted. This report is such an example.

Here the authors report an analysis of the epidemiology of RSV in Thailand using retrospective Hospitalization data collected in a Universe Care Scheme database. There are a few issues to address.

Major:

  1. The authors somehow ignored the potential of the COVID-19 pandemic. In their analysis, the year 2020 is separated from other years, but the year count here starts in October. So the Fiscal Year 2019 (FY2019) includes months of the pandemic. In fact, the FY2019 December to March figure 1 has more cases, for the period, than any previous FY (same periods). So I would suggest excluding the months of December to March from FY2019 in the figure 2 analysis. And discuss this.

  2. A complex issue of mortality analysis is the ability to separate the severity of the disease at admission from the overall care management of the health care system. So, can you provide, in addition to current mortality, the mortality within the first 48 hours of hospitalization and mortality post-48 hours of hospitalization? Did this change with the pandemic?

  3. You are using a special ICD10 classification. It is modified for Thailand. A comment on how this is comparable internationally is needed.

Minor:

  1. Statistical analysis section:
    - Please write Stata not STATA. Stata is not an acronym as SPSS or SAS. See the official documentation please to confirm.
    - Line 98 states to use population. Please indicate the source of the population estimates.
    - Line 100 mentions person-year. Please, indicate how the person-time is calculated.

  2. Figure 2 Please write FY2015-2019 and FY2020 to alert the readers that the years count differently. And please, see my previous comment about the pandemic months in FY2019.
    - Also, clarify how the pooled FY2015-2019 rates were computed.

  3. Table 1
    - Line 122, Please, say more than just "Demographic data".

  4. Table 2:
    - Report, the person-time
    - We do not need the p-value here. In fact, except for the association assessment, no other part of this report there is any p-value. If you want p-value here document how it is computed, report for admissions as well, and report p-values on table 3.

  5. Table 3:
    - Could be merged to table 2 and mortality measures added.

  6. Lines 163/164 - it is written "The rate of endotracheal intubation was higher in children aged 1 to 5 years (398/1331, 29.9%) than in children aged 5 to 10 years (42/1331; 3.16%)". This is incorrect. The word "rate" should be replaced by "fraction". If you want to compare rates please compare the rates in person-time for each age group. Please be clearer on what measure you are using.

  7. The time here is counted as Thailand Fiscal Year. It causes confusion the text when the year is mentioned. Please can you prefix the year with "FY"? Example FY2020 rather than just 2020.

  8. Line 55 - what are the percentages in the brackets mean? Please, clarify.

  9. Line 61 - when the authors say "in a previous study". Please, cite the referred study.

  10. Line 96 - please correct STATA to Stata. Stata is not an acronym (see the official Stata documentation).

  11. Tables 4 and 5: Please, replace "multivariate" by "multivariable".

  12. Tables 4 and 5:
    - These tables include OR. And nowhere in the results, these measures are used in the results
    - Why the use of OR? OR exaggerates the association and it is hard to many readers to understand. I suggest the authors to use relative risk (RR).
    - Show the number of observations (numerator and denominator). In the future, this report can be included in a meta-analysis. Such results could be used.

  13. Lines 200/201 - the terminology "non-pharmaceutical treatments" is imprecise. Please use an expression more towards preventive measures.

Author Response

Dear Editors and Reviewer

Thank you for taking your time reviewing our manuscript. Your comments are very valuable for improving our writing. We are glad to receive all your valuable comments. Therefore, the authors have discussed, looked back, and edited the manuscript according to the constructive feedback of the manuscript. We really hope that our revision will match the criteria for publication in Children. Our responses to editors are described as follow.

Reviewer # 2

Globally, in South Global, there have been major reductions in bacterial respiratory infection in under-5, so it became clear how urgent the investments in the prevention, diagnosis and management of children's respiratory infections are. So surveillance of these viral diseases is needed, and regular and timely analysis should be conducted. This report is such an example.

Here the authors report an analysis of the epidemiology of RSV in Thailand using retrospective Hospitalization data collected in a Universe Care Scheme database. There are a few issues to address.

Major:

The authors somehow ignored the potential of the COVID-19 pandemic. In their analysis, the year 2020 is separated from other years, but the year count here starts in October. So the Fiscal Year 2019 (FY2019) includes months of the pandemic. In fact, the FY2019 December to March figure 1 has more cases, for the period, than any previous FY (same periods). So I would suggest excluding the months of December to March from FY2019 in the figure 2 analysis. And discuss this.

Response: Thank you for your feedback. The fiscal year 2020 began in October 2019 and ended in September 2020. As a result, the fiscal year 2019 did not include any months affected by the epidemic. The COVID-19 pandemic and lockdown in Thailand began in April 2020. As we discussed in the discussion section, we discovered that the rate of RSV-related LRTI hospitalization decreased considerably. In addition, we define fiscal year in methods.

A complex issue of mortality analysis is the ability to separate the severity of the disease at admission from the overall care management of the health care system. So, can you provide, in addition to current mortality, the mortality within the first 48 hours of hospitalization and mortality post-48 hours of hospitalization? Did this change with the pandemic?

Response: Thank you for your feedback. However, we are unable to offer present mortality, mortality within the first 48 hours of hospitalization, or mortality after the first 48 hours of hospitalization, which we recognize as a limitation of our study.

You are using a special ICD10 classification. It is modified for Thailand. A comment on how this is comparable internationally is needed.

Response:  Despite the fact that we utilize ICD-10 TM, the WHO classification of ICD-10 is J20.5 RSV bronchitis, J21.0 RSV bronchiolitis, and J12.1 RSV pneumonia, which is the same code as ICD-10 TM.

Minor:

Statistical analysis section:

- Please write Stata not STATA. Stata is not an acronym as SPSS or SAS. See the official documentation please to confirm.

Response: We much appreciate your feedback. We have changed the word to “Stata” as shown in line 101.

- Line 98 states to use population. Please indicate the source of the population estimates.

Response: We much appreciate your feedback. In addition, “the population estimate data was extracted from official statistic registration systems, The Bureau of Registration Administration, Department of Provincial Administration, Ministry of Interior, Thailand.” is added in line 104-106.

- Line 100 mentions person-year. Please, indicate how the person-time is calculated.

Response: Thank you very much for your feedback. The mortality rate of RSV related LRTI in our study was calculated from number of deaths attributed to RSV LRTI divided with the size of the population at the midpoint of the Fiscal year 2015-2020 and expressed per 100,000 population of the same age groups. We rewrite it to “The mortality rate was calculated and presented as per 100,000 population of the same age groups.” As shown on line 108-109.

Figure 2 Please write FY2015-2019 and FY2020 to alert the readers that the years count differently. And please, see my previous comment about the pandemic months in FY2019.

Response: As indicated in the preceding comments, we present the definition of fiscal years as part of the methods for the reader's better understanding.

- Also, clarify how the pooled FY2015-2019 rates were computed.

Response: As indicated in the preceding comments, we present the definition of fiscal years as part of the methods for the reader's better understanding.

Table 1

- Line 122, Please, say more than just "Demographic data".

Response: Thank you very much for your feedback. We rewrite it to “The proportion of RSV LRTI in hospitalized children during the fiscal years 2015–2020.” on name of table 1.

Table 2:

- Report, the person-time

Response: Thank you very much for your feedback. In addition, we had changed “person-year” into “person-time” in Table 2.

- We do not need the p-value here. In fact, except for the association assessment, no other part of this report there is any p-value. If you want p-value here document how it is computed, report for admissions as well, and report p-values on table 3.

Response: Thank you very much for your feedback. Furthermore, we removed the p-value from table 2.

Table 3:

- Could be merged to table 2 and mortality measures added.

Response: Thank you very much for your feedback. We can't merge since Table 3 shows admission trends by year, but Table 2 shows admission and mortality trends by age group.

Lines 163/164 - it is written "The rate of endotracheal intubation was higher in children aged 1 to 5 years (398/1331, 29.9%) than in children aged 5 to 10 years (42/1331; 3.16%)". This is incorrect. The word "rate" should be replaced by "fraction". If you want to compare rates please compare the rates in person-time for each age group. Please be clearer on what measure you are using.

Response: Thank you very much for your feedback. We rewrite it to “fraction” as shown on line 173.

The time here is counted as Thailand Fiscal Year. It causes confusion the text when the year is mentioned. Please can you prefix the year with "FY"? Example FY2020 rather than just 2020.

Response: Thank you very much for your feedback. We used prefix Fiscal year when the year is mentioned.

Line 55 - what are the percentages in the brackets mean? Please, clarify.

Response: Thank you very much for your feedback. The percentages in the brackets represent prevalence of pathogen causing LRTI among children in that study.

Line 61 - when the authors say "in a previous study". Please, cite the referred study.

Response: Thank you very much for your feedback. We had added the reference [8] as shown on line 66.

Line 96 - please correct STATA to Stata. Stata is not an acronym (see the official Stata documentation).

Response: Thank you very much for your feedback. We rewrite it to “Stata”.

Tables 4 and 5: Please, replace "multivariate" by "multivariable".

Response: Thank you very much for your feedback. We rewrite it to “multivariable” as shown in Tables 4 and 5.

Tables 4 and 5:

- These tables include OR. And nowhere in the results, these measures are used in the results

Response: Thank you very much for your feedback. We added the value of odds ratio, “Chronic respiratory disease (adjusted odds ratio (aOR) 1.67; 95% CI 1.20-2.32, P = 0.002), congenital heart disease (aOR 5.19; 95% CI 4.22-6.40, P <0.001), malnutrition (aOR 3.02; 95% CI 1.99-4.59, P <0.001), and neurological disease (aOR 2.57; 95% CI 1.55-4.29, P <0.001) were the most prevalent comorbidities of respiratory failure requiring endotracheal intubation among children with RSV-related LRTI (Table 4).” as shown on line 175-179, and “Factors associated with death were related to underlying medical conditions, including congenital heart disease (aOR 10.09; 95% CI 6.21-16.37, P <0.001), hematologic malignancy (aOR 10.34; 95% CI 4.17-25.63, P <0.001), and neurological disease (aOR 3.50; 95% CI 1.13-10.89, P =0.030) (Table 5).” as shown on line 189-192.

- Why the use of OR? OR exaggerates the association and it is hard to many readers to understand. I suggest the authors to use relative risk (RR).

Response: Thank you very much for your feedback. Our secondary objective was to assess the risk factors associated with severe RSV LRTI-related intubation and mortality. Odds ratio is a measure of association between an exposure and an outcome. Because the odds ratio can determine whether a particular exposure is a risk factor for an outcome and compare the magnitude of various risk factors for that outcome. Odds ratios are commonly used in case-control studies; however, they can also be used in cross-sectional and cohort study designs as well. Therefore, we used odds ratio to assess the risk factors of intubation and mortality in our study.

- Show the number of observations (numerator and denominator). In the future, this report can be included in a meta-analysis. Such results could be used.

Response: As recommended, we add the number of observations to tables 4 and 5.

Lines 200/201 - the terminology "non-pharmaceutical treatments" is imprecise. Please use an expression more towards preventive measures.

Response: Thank you very much for your valuable suggestion. We rewrite the sentence to “This may have been the result of the wide adoption of public health policies following the start of the pandemic, including the use of non-pharmaceutical treatments (NPIs), such as the use of face masks, hand washing, social distancing, and school closures [10].” as shown in line 212-215.

Reviewer 3 Report

In this paper, the authors present a retrospective analysis of the epidemiology, seasonal variations, morbidity, and mortality of respiratory syncytial virus (RSV), other than an assessment of factors involved in the worsening of RSV-related lower respiratory tract infections (LRTI).

The fundamental issue with this paper is that it requires considerable linguistic revision.

The paper technically sounds. The statistical analyses seem to be well-conducted with the usage of proper tests. I have several comments and suggestions for the authors that need to be addressed.

Abstract

Revise punctuation (e.g. “%” instead of “percent”, is sounds clearer)

- “person-years” instead of “per-son-years”

Introduction

In general, the introduction needs to be more exhaustive and argumentative. Moreover:

- “Respiratory syncytial virus (RSV) is the most common cause of bronchiolitis in infants and young children.” This affirmation needs a reference which considers a relevant number of evidence. I suggest: Florin TA, Plint AC, Zorc JJ. Viral bronchiolitis. Lancet. 2017;389(10065):211-224. doi:10.1016/S0140-6736(16)30951-5

- Second period needs a reference, too.

- “As a result of this surveillance, pathogens such as influenza (8.2%)”. The pathogen is the influenzavirus.

- “organism-specific” instead of “organism specific”.

- “and so the incidence of this infection has been underreported in a previous study”. Try to rephrase.

- At the end of the introduction, write something about the secondary objective you cited in the abstract, which is the identification of risk factors.

Materials and Methods

Study design and participants

- “RSV will be diagnosed”. Why the use of a future tense?

- Why didn’t you use data derived from fiscal year 2021? Aren’t they available yet? It could have been interesting to investigate the intrapandemic period.

Statistical analyses

- “All statistical analyses were performed using the STATA software version 10 (StataCorp LP)”: please, provide a reference. 

- Regarding the multivariable logistic regression, which method did you use? It was not a full model; was it a stepwise selection? A backward, a forward one? Please explain.

Results

The results section needs to be revised in terms of grammar clarity and correctness. In particular:

- “The most prevalent diagnosis” instead of “The most prevalence diagnosis”.

- “followed” instead of “follow”.

- “The highest hospitalization from RSV-related LRTI was detected in tertiary care hospitals”: it is not clear what the authors meant to say.

- “The detail of demographic data divided by principal diagnosis was presented in Table 1”: rewrite it in a clearer way.

- “Among the children hospitalized with RSV LRTI, 49.54% (9581/19,340) were aged 1 to under 5 years, 48.27% (9,335/19,340) were aged under 1 years, and 2.19% were aged 5 to under 18 years (424/19,340)”: the use of “were aged” is redundant, it is general advice to change the expressions used (e.g. “were under 1 year of age” instead of “were aged under 1 years”). Moreover, it’s “under 1 year”.

- “The RSV-related LRTI incidence in infants under 1-year-126 old, 1 to under 5, and 5 to under 18 years were accounted for 2.60, 0.58, and 0.01 per 1,000 127 person-year, consecutively (Table 2)”: “respectively” instead of “consecutively”. Moreover, present Table 2 more properly, rewrite this sentence.

- “The annual admission number from RSV-related LRTI in children was shown in Table 3”: as stated in the previous bullet point, rewrite this sentence to better announce Table 3.

- “Incidence of […] increased” instead of “was increasing”.

- “It took the lowest hospitalization during summer of Thailand (March to May)”: it is not clear what the authors wrote.

- “When compare admission rate of fiscal year 2015 to 2019 with fiscal year 2020”: rephrase this sentence. 

- “Nevertheless, the admission rate was lower compare with the previous year in the same month since April 2020 (Figure 2)”: also here, it needs to be rewritten. Moreover, I believe it is important to notice that from the month of April 2020, admission rates were sensitively lower when compared to the 2015-2019 period. It needs to be added, since it may be an effect of the Covid-19 pandemic.

- “Length of stay in hospitalized children with RSV-related LRTI difference in diagnosis”: what did the authors mean to say? It is not even a sentence since the verb is missing. Moreover, the name of the paragraph needs to be modified.

- “It was most prevalent in children with RSV pneumonia, which accounted for 1,257 cases (94.44%)”: this result needs to be better explained. I suggest connecting it with the previous sentence, since it is a subgroup.

- “Under one-year-old children accounted for […]”: rephrase it.

- “Mortality rate” section needs to be rewritten, too.

Tables and Figures captions need to be rewritten.

Discussion

The Discussion seems to be rich in content, maybe too much. I recommend moving some parts of it to the Introduction section (RSV clinical manifestations, for example). Moreover:

- “the primary social health insurance system in Thailand (UC scheme) […] covers roughly 72 percent of the population”: what about the other 28% of the population? Which part of the population is not covered? It needs to be stated and added in the possible limits of the study, since it is needed to know, for example, if they have a lower socioeconomic status than the 72% of the population, which is covered.

- “varies greatly among countries” instead of “varies greatly from nation to country”.

- “This may result from the non-pharmaceutical treatments (NPIs) performed by public health”: I would say that “This may be the result of Public Health policies widely adopted, which comprised the use of non-pharmaceutical treatments (NPIs)”. Moreover, I suggest searching for other references.

- “RSV seasons are varied around the world, it depends on altitude, climate, and across geographic locations”: rephrase it.

- “the late rainy months” instead of “[…] month”.

- “It can explain from temperature and humidity are influence the epidemic pattern of RSV”: this is not written in English.

- “death rate” should be modified: “mortality rate” or “fatality rate”, in particular. It depends on what you are referring to.

- “this was a retrospective reviewed the diagnosis of hospitalization from ICD-10-TM. It may cause the underestimation of RSV-related LRTI if the coder recorded incorrected diagnosis”: rephrase it, for example “it was a retrospective review of the records derived from the ICD-10-TM diagnosis classification: this may have caused the underestimation of RSV-related LRTI, since there could have been some coder diagnosis misclassifications”.

- “Possible risk factor for the development of RSV LRTI […]” instead of “The risk for develops RSV LRTI […]”.

Conclusion

- “Effective preventative measures should target children with these risk factors in order to lower the likelihood of hospitalization and serious illness”: I would say that effective preventive measures should target primarily children with risk factor; it’s a matter of public health priorities.

Author Response

Dear Editors and Reviewer

Thank you for taking your time reviewing our manuscript. Your comments are very valuable for improving our writing. We are glad to receive all your valuable comments. Therefore, the authors have discussed, looked back, and edited the manuscript according to the constructive feedback of the manuscript. We really hope that our revision will match the criteria for publication in Children. Our responses to editors are described as follow.

Reviewer # 3

In this paper, the authors present a retrospective analysis of the epidemiology, seasonal variations, morbidity, and mortality of respiratory syncytial virus (RSV), other than an assessment of factors involved in the worsening of RSV-related lower respiratory tract infections (LRTI).

The fundamental issue with this paper is that it requires considerable linguistic revision.

The paper technically sounds. The statistical analyses seem to be well-conducted with the usage of proper tests. I have several comments and suggestions for the authors that need to be addressed.

Abstract

Revise punctuation (e.g. “%” instead of “percent”, is sounds clearer)

Response: Thank you very much for your valuable suggestion. We rewrite to “%” as recommended.

- “person-years” instead of “per-son-years”

Response: Thank you very much for your valuable suggestion. We rewrite to “person-years” as recommended.

Introduction

In general, the introduction needs to be more exhaustive and argumentative. Moreover:

- “Respiratory syncytial virus (RSV) is the most common cause of bronchiolitis in infants and young children.” This affirmation needs a reference which considers a relevant number of evidence. I suggest: Florin TA, Plint AC, Zorc JJ. Viral bronchiolitis. Lancet. 2017;389(10065):211-224. doi:10.1016/S0140-6736(16)30951-5

Response: Thank you very much for your valuable suggestion. We already changed the reference for that sentence as your suggestion in line 46.

- Second period needs a reference, too.

Response: Thank you very much for your valuable suggestion. We already added the reference for that sentence as your suggestion in line 51.  

- “As a result of this surveillance, pathogens such as influenza (8.2%)”. The pathogen is the influenzavirus.

Response: Thank you very much for your valuable suggestion. We rewrite to “influenza virus” as shown in line 58.

- “organism-specific” instead of “organism specific”.

Response: Thank you very much for your valuable suggestion. We rewrite to “organism-specific” as shown in line 61.

- “and so the incidence of this infection has been underreported in a previous study”. Try to rephrase.

Response: Thank you very much for your valuable suggestion. This revision we put out for expert English editing service of MDPI as the certification stated above. And rephrase to “and so the incidence of this infection was underreported in a previous study” in line 63-64.

- At the end of the introduction, write something about the secondary objective you cited in the abstract, which is the identification of risk factors.

Response: Thank you very much for your valuable suggestion. In addition, “we assessed the risk factors associated with severe RSV LRTI-related morbidity and mortality.” is added at the end of the introduction.

Materials and Methods

Study design and participants

- “RSV will be diagnosed”. Why the use of a future tense?

Response: Thank you very much for your feedback. In addition, “RSV was diagnosed as a cause of LRTI when respiratory tract specimens were detected from molecular diagnosis” is added on line 81-82.

- Why didn’t you use data derived from fiscal year 2021? Aren’t they available yet? It could have been interesting to investigate the intrapandemic period.

Response: Thank you for your valuable feedback; unfortunately, fiscal year 2021 data is unavailable.

Statistical analyses

- “All statistical analyses were performed using the STATA software version 10 (StataCorp LP)”: please, provide a reference.

Response: Thank you very much for your feedback. We add the suggested citation for the Stata 10 software  “(StataCorp. 2007. Stata Statistical Software: Release 10. College Station, TX: StataCorp LP.)” as shown on line 101-102.

- Regarding the multivariable logistic regression, which method did you use? It was not a full model; was it a stepwise selection? A backward, a forward one? Please explain.

 Response: Thank you very much for your feedback. In addition, “The univariate and backward stepwise multivariable logistic regression analysis was applied to determine the factors associated with respiratory failure that required endotracheal intubation and mortality.” is added in Line 111.

Results

The results section needs to be revised in terms of grammar clarity and correctness. In particular:

- “The most prevalent diagnosis” instead of “The most prevalence diagnosis”.

Response: Thank you very much for your feedback. We rewrite it to “The most prevalent diagnosis” as shown on line 123-124.

- “followed” instead of “follow”.

Response: Thank you very much for your valuable suggestion. We rewrite it to “followed” as shown on line 128.

- “The highest hospitalization from RSV-related LRTI was detected in tertiary care hospitals”: it is not clear what the authors meant to say.

Response: Thank you very much for your valuable suggestion. This revision we put out for expert English editing service of MDPI as the certification stated above.  “Tertiary care hospitals had the highest rate of RSV-related LRTI hospitalization (10,620/19,340; 54.91%), followed by secondary (27.26%), private (11.66%), and primary care hospitals (6.17%). The proportion of RSV LRTI in hospitalized children divided by principal diagnosis is presented in Table 1.”

- “The detail of demographic data divided by principal diagnosis was presented in Table 1”: rewrite it in a clearer way.

Response: Thank you very much for your valuable suggestion. We rewrite it to “The proportion of RSV LRTI in hospitalized children divided by principal diagnosis was presented in Table 1”

- “Among the children hospitalized with RSV LRTI, 49.54% (9581/19,340) were aged 1 to under 5 years, 48.27% (9,335/19,340) were aged under 1 years, and 2.19% were aged 5 to under 18 years (424/19,340)”: the use of “were aged” is redundant, it is general advice to change the expressions used (e.g. “were under 1 year of age” instead of “were aged under 1 years”). Moreover, it’s “under 1 year”.

Response: Thank you very much for your valuable suggestion. We rewrite it to “were under 1 year of age” as shown on line 134.

- “The RSV-related LRTI incidence in infants under 1-year-126 old, 1 to under 5, and 5 to under 18 years were accounted for 2.60, 0.58, and 0.01 per 1,000 127 person-year, consecutively (Table 2)”: “respectively” instead of “consecutively”. Moreover, present Table 2 more properly, rewrite this sentence.

Response: Thank you very much for your valuable suggestion. We rewrite it to “respectively” as shown on line 137. This revision we put out for expert English editing service of MDPI as the certification stated above.

- “The annual admission number from RSV-related LRTI in children was shown in Table 3”: as stated in the previous bullet point, rewrite this sentence to better announce Table 3.

Response: Thank you very much for your valuable suggestion. We added “The annual admission number of children due to RSV-related LRTI from fiscal years 2015 to 2020 is shown in Table 3.” In line 141-142.

- “Incidence of […] increased” instead of “was increasing”.

Response: Thank you very much for your valuable suggestion. We rewrite it to “increased” as shown on line 145.

- “It took the lowest hospitalization during summer of Thailand (March to May)”: it is not clear what the authors wrote.

Response: Thank you very much for your feedback. In addition, “The lowest hospitalization happened during summer of Thailand (March to May).” as shown in line 148-149.

- “When compare admission rate of fiscal year 2015 to 2019 with fiscal year 2020”: rephrase this sentence.

Response: Thank you very much for your valuable suggestion. This revision we put out for expert English editing service of MDPI as the certification stated above. And rephrase to “In comparing the admission rate during fiscal years 2015 to 2019 to that during fiscal year 2020” in line 156-157.

- “Nevertheless, the admission rate was lower compare with the previous year in the same month since April 2020 (Figure 2)”: also here, it needs to be rewritten. Moreover, I believe it is important to notice that from the month of April 2020, admission rates were sensitively lower when compared to the 2015-2019 period. It needs to be added, since it may be an effect of the Covid-19 pandemic.

Response: Thank you very much for your feedback. We rewrite the sentence to “In comparing the admission rate during fiscal years 2015 to 2019 to that during fiscal year 2020, the admission rate in October 2019 to March 2020 was higher than that in fiscal years 2015 to 2019. Nevertheless, the admission rate from April to September 2020 was lower compared with this same period in fiscal years 2015 to 2019 (Figure 2).” as shown in line 156-159.

- “Length of stay in hospitalized children with RSV-related LRTI difference in diagnosis”: what did the authors mean to say? It is not even a sentence since the verb is missing. Moreover, the name of the paragraph needs to be modified.

Response: Thank you very much for your feedback. We rewrite the sentence to “Each diagnosis of RSV-related LRTI in hospitalized children led to a different length of stay. RSV pneumonia accounted for the longest length of stay (7.9 ± 18.5 days), followed by RSV bronchitis (4.4 ± 5.8 days) and RSV bronchiolitis (4.1 ± 4.3 days), consecutively.” as shown in line 165-167.

- “It was most prevalent in children with RSV pneumonia, which accounted for 1,257 cases (94.44%)”: this result needs to be better explained. I suggest connecting it with the previous sentence, since it is a subgroup.

Response: Thank you very much for your valuable suggestion. This revision we put out for expert English editing service of MDPI as the certification stated above. And revised to “There were 1,331 cases of acute respiratory failure necessitating intubation out of a total of 19,340 RSV LRTIs (6.88%), with the highest prevalence occurring in children with RSV pneumonia, which accounted for 1,257 cases (94.44%)” in line 169-171.

- “Under one-year-old children accounted for […]”: rephrase it.

Response: Thank you very much for your valuable suggestion. This revision we put out for expert English editing service of MDPI as the certification stated above.

- “Mortality rate” section needs to be rewritten, too.

Response: Thank you very much for your feedback. We rewrite it to “RSV-related LRTI mortality”

Tables and Figures captions need to be rewritten.

Response: Thank you very much for your valuable suggestion. This revision we put out for expert English editing service of MDPI and revised all captions as recommended.

Discussion

The Discussion seems to be rich in content, maybe too much. I recommend moving some parts of it to the Introduction section (RSV clinical manifestations, for example). Moreover:

Response: Thank you for your insightful comments. We omitted certain sentences from the discussion that were mentioned in the introduction. “The most recent global burden research anticipated that 33 million RSV LRTIs would occur in children aged 0 to 60 months worldwide in 2019, and that 3.6 million of these children would require hospitalization [10]. The earlier national statistics of LRTI among children in Thailand from 2010 [8] indicated hospital admissions owing to LRTI in children, however RSV was not included as an etiologic agent of LRTI due to the limitations of laboratory investigation”

- “the primary social health insurance system in Thailand (UC scheme) […] covers roughly 72 percent of the population”: what about the other 28% of the population? Which part of the population is not covered? It needs to be stated and added in the possible limits of the study, since it is needed to know, for example, if they have a lower socioeconomic status than the 72% of the population, which is covered.

Response: Thank you very much for your valuable comments. We include this as a limitation of our study. “Fifth, the data for the analysis were derived from the public health ministry's universal coverage program, which covers 47.8 million people, or 72 percent of the population. However, we do not have statistics on the remaining 28%, which is fully reimbursable through the government system (finance ministry and labor ministry) [9].”

- “varies greatly among countries” instead of “varies greatly from nation to country”.

Response: Thank you very much for your feedback. We rewrite the sentence to “varies greatly among countries” as shown in line 202.

- “This may result from the non-pharmaceutical treatments (NPIs) performed by public health”: I would say that “This may be the result of Public Health policies widely adopted, which comprised the use of non-pharmaceutical treatments (NPIs)”. Moreover, I suggest searching for other references.

Response: Thank you very much for your feedback. We rewrite the sentence to “This may have been the result of the wide adoption of public health policies following the start of the pandemic, including the use of non-pharmaceutical treatments (NPIs), such as the use of face masks, hand washing, social distancing, and school closures [10].” as shown in line 212-215.

- “RSV seasons are varied around the world, it depends on altitude, climate, and across geographic locations”: rephrase it.

Response: Thank you very much for your valuable suggestion. This revision we put out for expert English editing service of MDPI as the certification stated above. And revised this sentence to “RSV seasons are varied around the world, depending on altitude, climate, and geographic location [16].” In line 230-231.

- “the late rainy months” instead of “[…] month”.

Response: Thank you very much for your feedback. We rewrite the sentence to “the late rainy months” as shown in line 232.

- “It can explain from temperature and humidity are influence the epidemic pattern of RSV”: this is not written in English.

Response: Thank you very much for your valuable suggestion. We rewrite the sentence to “It can explain from changing in human behavior during rainfall season. The children spend more time indoor during stay with family and in school. This crowding environment could influence the RSV transmission, as RSV transmit through direct and indirect contact [19]. The wet conditions during rainy seasons also increasing the amount of virus that is deposited on surfaces, and virus survival in droplets on surfaces which encourage contact transmission of RSV [20].” as shown in line 286-291.

-  “death rate” should be modified: “mortality rate” or “fatality rate”, in particular. It depends on what you are referring to.

Response: Thank you very much for your valuable suggestion. We modified “death rate” to “mortality rate”

- “this was a retrospective reviewed the diagnosis of hospitalization from ICD-10-TM. It may cause the underestimation of RSV-related LRTI if the coder recorded incorrected diagnosis”: rephrase it, for example “it was a retrospective review of the records derived from the ICD-10-TM diagnosis classification: this may have caused the underestimation of RSV-related LRTI, since there could have been some coder diagnosis misclassifications”.

Response:  Thank you very much for your valuable suggestion. We rewrite the sentence to “it was a retrospective review of the records derived from the ICD-10-TM diagnosis classification, which may have caused an underestimation of RSV-related LRTI, since there could have been some coder diagnosis misclassifications.” as shown in line 271-274.

- “Possible risk factor for the development of RSV LRTI […]” instead of “The risk for develops RSV LRTI […]”.

Response: Thank you very much for your valuable suggestion. We rewrite the sentence to “Possible risk factors for the development of RSV LRTI” as shown in line 279-280.

Conclusion

- “Effective preventative measures should target children with these risk factors in order to lower the likelihood of hospitalization and serious illness”: I would say that effective preventive measures should target primarily children with risk factor; it’s a matter of public health priorities.

Response: Thank you very much for your valuable suggestion. We rewrite the sentence to “Effective preventive measures should primarily target children with risk factors; it is a matter of public health priorities.” as shown in line 306-307.

Reviewer 4 Report

The article was well written and organized. However, the following suggestions will help improve it:

1. Lines 42: Pls give a brief summary of the virology of RSV for example; viral family, its pathogenesis, and virulence factors to improve the article scientific soundness.

2. Lines 46: Explain the reasons why children below the age of 5 are most vulnerable to RSV and cite relevant studies.

3. Lies 115 in the result section, please include common treatment/medications given to Thai children with RSV/management of RSV patients since you already talked about treatment outcomes in lines 154.

4. Line 278 in the conclusion section; include other possible factors that could worsen clinical outcomes in children with RSV infections, such as immature immune system, genetic diseases (Cystic fibrosis), Upper respiratory tract malformation (Choanal atresia, Pyriform aperture stenosis, etc).

Author Response

Dear Editors and Reviewer

Thank you for taking your time reviewing our manuscript. Your comments are very valuable for improving our writing. We are glad to receive all your valuable comments. Therefore, the authors have discussed, looked back, and edited the manuscript according to the constructive feedback of the manuscript. We really hope that our revision will match the criteria for publication in Children. Our responses to editors are described as follow.

Reviewer # 4

The article was well written and organized. However, the following suggestions will help improve it:

  1. Lines 42: Pls give a brief summary of the virology of RSV for example; viral family, its pathogenesis, and virulence factors to improve the article scientific soundness.

Response: Thank you very much for your valuable suggestion. We add the sentence “Respiratory syncytial virus (RSV) is the type species of Genus Pneumovirus, Sub-family Pneumovirinae, Family Paramyxoviridae, Order Mononegavirales. Human RSV exists as two antigenic subgroups, A and B” as shown in line 43-45.

  1. Lines 46: Explain the reasons why children below the age of 5 are most vulnerable to RSV and cite relevant studies.

Response: Thank you for your insightful comments. We cannot discover compelling evidence to support this statement. We'd like to change this sentence to “with children under the age of 6 months account for 45% of hospital admissions and mortality [4]” as shown in line 47-48.

  1. Lies 115 in the result section, please include common treatment/medications given to Thai children with RSV/management of RSV patients since you already talked about treatment outcomes in lines 154.

Response: Thank you very much for your valuable comment. The length of stay and the requirement for an endotracheal tube are the primary treatment outcomes. Unfortunately, we did not have any data on regularly used medications and treatments from the data registration system.

  1. Line 278 in the conclusion section; include other possible factors that could worsen clinical outcomes in children with RSV infections, such as immature immune system, genetic diseases (Cystic fibrosis), Upper respiratory tract malformation (Choanal atresia, Pyriform aperture stenosis, etc).

Response: Thank you very much for your insightful comments. We added “as well as potential factors that may aggravate clinical outcomes in children with RSV infections, such as an immature immune system, hereditary disorders (cystic fibrosis), and upper respiratory tract deformity (choanal atresia, pyriform aperture stenosis, etc.).” in the conclusion section line 303-306 as recommended.

Round 2

Reviewer 3 Report

The Authors' manuscript has substantially increased in quality.

I consider the existing form suitable for publication.